# Research Progresses on the Function and Detection Methods of Insect Gut Microbes

**DOI:** 10.3390/microorganisms11051208

**Published:** 2023-05-05

**Authors:** Yazi Li, Liyun Chang, Ke Xu, Shuhong Zhang, Fengju Gao, Yongshan Fan

**Affiliations:** Tangshan Key Laboratory of Agricultural Pathogenic Fungi and Toxins, Department of Life Science, Tangshan Normal University, Tangshan 063000, China

**Keywords:** insect, gut flora, gut microbial function, detection method

## Abstract

The insect gut is home to an extensive array of microbes that play a crucial role in the digestion and absorption of nutrients, as well as in the protection against pathogenic microorganisms. The variety of these gut microbes is impacted by factors such as age, diet, pesticides, antibiotics, sex, and caste. Increasing evidence indicates that disturbances in the gut microbiota can lead to compromised insect health, and that its diversity has a far-reaching impact on the host’s health. In recent years, the use of molecular biology techniques to conduct rapid, qualitative, and quantitative research on the host intestinal microbial diversity has become a major focus, thanks to the advancement of metagenomics and bioinformatics technologies. This paper reviews the main functions, influencing factors, and detection methods of insect gut microbes, in order to provide a reference and theoretical basis for better research utilization of gut microbes and management of harmful insects.

## 1. Introduction

The insect microbial community plays an important role in nutrient absorption, development of immunity, and resistance to foreign pathogenic organisms [1,2,3]. Gut microbes and hosts are interdependent and mutually restrictive and play an important role in host health [4]. Alterations in the intestinal community may influence insecticide resistance, physiological functions, and the eating habits of the host. Insect intestinal microorganisms have shown great application potential in the fields of pest prevention and control, utilization of insect resources, and industrial production (such as degrading cellulose and providing trace elements) [5,6,7]. Currently, functional studies of insect intestinal microorganisms remain challenging, in part because of the complexity of their composition, which may vary widely among individuals [8,9]. Therefore, understanding the function and composition of insect intestinal microorganisms is very important for regulating the rise and fall of the population, studying the relevant evolutionary mechanisms, and excavating strains with special functions [7].

Historically, traditional culture-based methods have been used to identify insect gut microbial diversity with many drawbacks (such as the fact that many intestinal bacteria cannot be obtained through culture and cannot systematically reveal the characteristics of insect intestinal flora), hindering the in-depth study of gut microbiota. In recent years, the development and application of new technologies such as metagenomics and bioinformatics have considerably enhanced our understanding of the composition and variety of the insect gut microbiota. This review summarizes insect gut function and the factors that influence insect gut microbes. In addition, we compared past and present methods for detecting insect intestinal microorganisms.

## 2. Functional Roles of the Insect Gut Microbiota

There are complicated microbial communities in the gut of insects, including bacteria, archaea, fungi, and viruses, which play an important role in the host’s nutrient metabolism, immune defense, and drug resistance enhancement. Insect gut microbiota supply host nutrition and maintain gut microbial homeostasis by generating antimicrobial compounds (bacteriocins and lipopeptides), metabolites such as short-chain fatty acids and vitamins, and by degrading plastics or inducing immune responses [3,10,11,12]. Xiao et al. found that *Drosophila* gut microbial homeostasis is synergistically regulated by Duox-ROS, AMPs, and C-type lectins [12]. In addition, the intestinal flora of insects can also recycle urea and uric acid, help insects defend against the infection of foreign pathogenic microorganisms, maintain body homeostasis, and facilitate the regular physiological activities of the host and the progress of various life activities [13].

### 2.1. Nutrient Metabolism

The intestinal microbiota of insects can secrete a variety of digestive enzymes to participate in the digestion of food and supply nutrients for insects. Common insects such as *Bombyx mori*, *Locusts*, *Aedes albopictus*, *Termite*, etc., have intestinal microorganisms that can secrete a variety of digestive enzymes. *Pseudomonas*, *Klebsiella pneumoniae*, and *Clostridium flexneri* in the intestinal microbiota of *B. mori* can secrete cellulase and have the ability to degrade carbohydrates [14]. Cellulase secreted by *Klebsiella pneumonia* in the intestinal tract of *Locusta migratoria manilensis* can degrade grass to generate carbohydrates, amino acids, and sugars for the host to utilize [15]. After *Aedes albopictus* ingests plant nectar, part of it is directly digested and absorbed, and the other part is transferred to the gut, where it is transformed into a common energy source for the host and gut microbes under the action of gut bacteria or fungi [16]. Cellobiohydrolase, which is secreted by microorganisms in the gut of *Termites*, can digest cellulose to provide energy for the host [17]. In addition, *Klebsiella*, *Proteus vulgaris*, *Erwinia* sp., and *Serratia liquefaciens* were isolated from the gut of *Diatraea saccharalis* larvae by in vitro culture, which can utilize starch, xylanolytic, pectinolytic, and polysaccharide, respectively [18]. Zheng et al. reveal that honeybee intestinal bacteria can degrade plant polymers from pollen and that the resulting metabolites provide nutrients for the host [19]. In addition, some intestinal symbionts can provide the host with amino acids and vitamins [20]. Hu et al. reported that the gut bacteria of ants utilize reclaimed recycled N to recycle urea or uric acid to synthesize amino acids required by the host in large quantities [13]. Duplais et al. reported that intestinal microorganisms in Herbivorous turtle ants *Cephalotes* can recycle metabolic waste rich in nitrogen to enrich nutritional components and supply amino acids for the host [21]. Blow et al. confirmed that *Acyrthosiphon pisum* can obtain vitamins B2 and B5 through intestinal symbiotic bacteria, *Buchnera aphidicola,* and supplement micronutrients [6]. These results indicate that gut microbes are a promising source of various digestive enzymes and trace elements.

In addition, the intestinal flora participates in the nutrient metabolism of insects and can also influence the feeding behavior, growth, and development of the host. Because of its simple intestinal microorganism species and easy feeding, *Drosophila* is often used as a model to study the function of symbiotic bacteria [22]. Wong et al. inoculated *Drosophila* eggs on a feed comprising *Acetobacter pomorum* and *Lactobacillus plantarum* (bacteria isolated from the gut or body of *Drosophila*) to investigate the behavioral responses of *Drosophila* to microorganisms. The research found that *Drosophila* vaccinated with *A. pomorum* had a diminished tendency to choose a high-protein diet, while flies vaccinated with *L. plantarum* were more likely to eat foods with higher carbohydrates [23]. In addition, both adults and larvae of *D. melanogaster* were attracted to volatile compounds from *Saccharomyces cerevisiae* and *L. plantarum* but repelled by *Acetobacter malorum* in behavioral assays [24]. Another study found that the germ-free *Rhynchophorus ferrugineus* Olivier larval development and weight had considerably diminished. In addition, after the introduction of intestinal microbiota, the content of hemolymph protein, glucose, and triglycerides significantly increased, and it was verified that the protein content level was restored after feeding sterile *R. ferrugineus* Olivier and *Lactococcus lactis*. *Enterobacter cloacae* significantly increases its hemolymph triglyceride and glucose content [25]. The changes in host behavior induced by these gut bacteria may reflect the metabolism of the gut microbiota and the nutritional needs of the host [26]. The above research supplies fundamental data for the breeding of insects and the functional research of intestinal microorganisms (Table 1).

### 2.2. Immune Defense

Insects are constantly in contact with pathogenic microorganisms such as viruses, fungi, and bacteria. Insects primarily rely on insect intestinal epithelial cells to generate reactive oxygen species (ROS) and antimicrobial peptides (AMPs) to resist the invasion of pathogenic microorganisms and preserve intestinal microbial homeostasis [27,28]. Recent research has shown that the insect gut microbiota plays a role in helping the host resist infection by foreign pathogens. Shao et al. found that *Enterococcus mundtii* can stably secrete a class IIa bacteriocin (mundticin KS) in the intestine of *Spodoptera littoralis* against invading bacteria to maintain the balance of the host gut microbial community [29]. Akbar et al. inoculated RPMI 1640 with intestinal bacteria isolated from cockroaches. Cockroach intestinal flora secretions can effectively inhibit Gram-positive (methicillin-resistant *Staphylococcus aureus*, *Streptococcus pyogenes*, *Bacillus cereus*) and Gram-negative (*Escherichia coli* K1, *Salmonella enterica*, etc.) [30]. Knight et al. isolated antifungal cyclic lipopeptides (bacteromycin F and fengycin) from the metabolites of *Bacillus subtilis*, which have inhibitory effects on a variety of fungi such as *Alternaria alternata*, *Aspergillus niger*, and *Cladosporium* sp. [31]. The above studies reveal that the metabolites of insect gut microbiota play an important role in maintaining gut microbial homeostasis and resisting the invasion of pathogenic microorganisms.

Recent studies have shown that gut commensals are important participants in the host immune system, and there are diverse interactions between gut commensals and the host immune system [32]. Pandey et al. analyzed the possible links between the gut microbial dynamics of the model organism *Spodoptera litura* and stress-inducing factors and found that *Pseudomonas* and *Enterobacter* are related to the inflammatory effects of insects. *Acinetobacter*, on the other hand, promotes larval fitness and decreases the inflammatory effects generated by dextran sulfate sodium [33]. Krams et al. demonstrated that *Enterococcus* was the main genus in the midgut of *G. mellonella* larvae, and the number of *Enterococcus* positively correlated with antimicrobial peptide-related genes (such as Gallerimycin, Gloverin, 6-tox, Cecropin-D, and Galiomicin) [1]. In addition, the *R. ferrugineus* Olivier larvae had higher antibacterial activity and phenoloxidase activity, while the immune-related genes and survival rate of sterile larvae were significantly down-regulated. The reintroduction of sterile *R. ferrugineus* Olivier larvae into the gut microbiota enhanced their immunity and survival [34]. Gao et al. revealed that the gut *PGRP-LA* gene of *Anopheles stephensi* could regulate immune responses by sensing the dynamics of the gut microbiota [35]. Park et al. demonstrated that the gelatinase secreted by *Enterococcus faecalis* in the intestinal tract of *Galleria mellonella* larvae can degrade antimicrobial peptides (Gm cecropin) in the hemolymph and damage the immune system [36]. The above studies have confirmed that intestinal commensal flora has a stimulating effect on host immunity. The research on the interaction mechanism between gut microbiota and insect immunity will help provide new strategies for pest management.

### 2.3. Antioxidation Function

Under normal circumstances, the normal metabolism of oxygen in the insect host will generate reactive oxygen and free radicals, but when they accumulate too much, they will destroy cells and endanger the health of the insect. Therefore, the elimination of excessive oxidation and free radicals can prevent related diseases. Previous studies indicated that different gut probiotics can exert their antioxidant power in different ways to maintain host health [37]. Saeedi et al. found that *Lactobacillus* can activate liver Nrf2 in *Drosophila* and mice by producing 5-methoxyindoleacetic acid, consequently achieving the “distant control” of oxidative stress in the liver and thus protecting the liver from oxidative damage induced by acetaminophen overdose and acute ethanol poisoning [38]. Elzeini et al. found that EPS (extracellular polysaccharide) generated by *Enterococcus faecalis*-HBE1, *Lactobacillus brevis*-HBE2, *E. faecalis*-HBE3 and *E. faecalis*-HBE4 isolated in the intestinal tract of *Apis mellifera* L. has antioxidant activity to scavenge DPPH free radicals [39]. Barretto et al. found that staphyloxanthin pigment generated by *Staphylococcus gallinarum* KX912244 isolated in the intestinal tract of *B. mori* can inhibit *S. aureus*, *E. coli,* and *Candida albicans* and scavenge DPPH free radicals [40]. Although the active metabolites of probiotics have better antioxidant functions, further investigation is essential to explore the mechanism of action.

### 2.4. Enhance Host Drug Resistance

Due to the excessive and irregular use of insecticides, insects have evolved drug resistance, which leads to non-target insect death and environmental pollution [41]. Insect drug resistance mechanisms are dominated by metabolic resistance and target resistance, and these mechanisms are primarily caused by the evolution of the host genome [41]. Studies have found that certain gut symbionts can degrade chemical pesticides, which in turn affects host resistance to pesticides. Chen et al. found that *Stegotrophomonas* of *B. mori* can increase the content of essential amino acids in the gut so that the larvae can more effectively avoid the effects of chlorpyrifos [42]. Trinder et al. found that supplementation of *Lactobacillus rhamnosus* in the *Drosophila melanogaster* diet could reduce the toxicity of organophosphorus pesticides [43]. Danilenko et al. described that *Lactobacillus* species of *L. plantarum* (strains H28, H24, KX519413, KX519414, LP8, LP25, LP86, LP95, and LP100) and *Lactobacillus kunkeei* were characterized by an increased antioxidant potential that protects against different pesticides in the gut of honey bees [44]. Sato et al. found that the intestinal symbiotic bacteria of *Riptortus pedestris* can degrade the organophosphorus insecticide into 3-methyl-4-nitrophenol, which is bactericidal but not insecticidal [45]. Itoh et al. found that the bed bug *R. pedestris* can acquire resistance to the insecticide phosphine (MEP) by acquiring MEP-degrading *Burkholderia* from the environment [46]. This indicates that the gut microbiota plays an important role in host drug resistance and that part of the gut microbiota can defend the host from chemical pesticides.

## 3. Factors Affecting the Gut Microbiome of Insects

### 3.1. Effects of Feed, Antibiotics and Culture Temperature on Insect Gut Microbes

The study finds food, antibiotics, and temperature have important effects on insect gut microbial diversity [47,48,49]. High-throughput sequencing technology was used to study the effects of *Spodoptera frugiperda* feeding on maize, wild oats, oilseed rape, and pepper on host gut microbial community structure and diversity. The results revealed that the gut microbial diversity of insects fed rapeseed was the highest and the gut microbial diversity of insects fed wild oats was the lowest, while the gut microbial diversity of insects fed maize without a seed coating agent was significantly higher than that with such an agent [50]. Liang et al. found that the midgut microbes of the 5th instar *Helicoverpa armigera* larvae feeding on lettuce leaves were significantly different from those of common silkworms at the genus level. The bacteria of the genera *Acinetobacter* and *Anaerofilum* are the main bacteria, while *Bacillus* and *Arcobacter* are the main bacteria in the normally fed *H. armigera* [51]. Priya et al. found that bacterial diversity varied widely between *H. armigera* from different host plants and the same host plant from different locations. Compared with insects that feed on crops, insects fed on artificial diets have significantly fewer gut microbial species [52]. Thakur et al. analyzed the effects of adding streptomycin to artificial diets on the survival and fitness of *Spodoptera litura* (Lepidoptera: Spodoptera) and its gut microbial diversity. Changes in microbial diversity were found in the guts of the larvae, with the larvae growing faster compared to the treatment group without antibiotics. The total activity of various digestive enzymes increased, and the activity of detoxification enzymes decreased significantly [48]. Wang et al. found that the overwintering stage of *Dendroctonus armandi* led to changes in the intestinal flora. The Proteobacteria (mainly γ-Proteobacteria) become the main phylum in the larva gut, followed by Actinobacteria and Firmicutes [53]. In addition, recent studies have found that the survival of *Nezara viridula* depends on intestinal symbiotic bacteria. An increase in temperature may result in a decrease in intestinal symbiotic bacteria content and host survival, and the antibiotic-treated animals also induced the same results [54]. Hence, studying the influence of the external environment on the microorganism diversity of the insect intestinal tract can be of positive significance for us to screen beneficial microorganisms and manage the rise and fall of pest populations [55].

### 3.2. Effects of Sex and Developmental Stage on Insect Gut Microbes

The diversity and proportion of insect gut microbes differed by sex and developmental period. The intestinal tract of male and female larvae of the 5th instar of *B. mori* was dominated by *Enterococcus*, *Delftia*, *Pelomonas*, *Ralstonia,* and *Staphylococcus*. The abundance of *Enterococcus* was significantly lower in female larvae than in male larvae, while the abundances of *Delftia*, *Aurantimonas*, and *Staphylococcus* were significantly increased [56]. Chen et al. reported that the dominant phyla Proteobacteria, Firmicutes, Actinobacteria, and Bacteroidetes were detected in the whole life history of silkworms and found that these four phyla were also present in the mulberry-eating larvae of *Acronicta major* and *Diaphania pyloalis*. The microbial community changes significantly between the early and late larvae of *B. mori*, consistent with host developmental changes [57]. The eggs of *Spodoptera exigua* are rich in *Enterococcus*, *Pseudomonas,* and *Asaia*, while *Methylobacter* and *Halomonas* are dominant in newly hatched larvae, and *Enterococcus* dominates in 3rd and 5th instar larvae. The pupal stage has the highest microbial diversity. There were no significant differences between newly hatched male and female *S. exigua* larvae [58]. In addition, bacteria and fungi showed dynamic changes at various developmental stages of the brown planthopper. The predominant fungal genus in the nymphal and adult stages was *Wallemia*, and the abundance of *Acinetobacter* in the egg stage was significantly reduced. The microbial community composition in female and male brown planthoppers is different and sex-dependent [59]. The above results reveal that there are different dominant flora in different developmental stages and sexes of insects, which may play an important role in different developmental stages and sexes, but the strains with specific functions need further analysis and experimental proof.

### 3.3. Effects of Pesticides on Insect Gut Microbes

Extensive use of pesticides will affect insects’ food digestion, nutrient absorption, metabolism, immune response, defense against pathogenic bacteria invasion, and microbial homeostasis [56,60,61]. Li et al. found that the activity and expression of enzymes related to nutrient metabolism in the midgut were unregulated, and the growth of *B. mori* was slow after feeding on phoxim [62]. Sun et al. studied the effects of microbial pesticides and camptothecin on the mortality of two lepidopteran insects, *Trichoplusia ni* and *S. exigua*. The bioassay results demonstrated that camptothecin significantly enhanced the toxicity of *B. thuringiensis* to *S. exigua* and *T. ni*, as well as to Autographa californica nucleopolyhedrovirus (AcMNPV) and *S. exigua* nuclear polyhedrosis virus (SeMNPV). It is speculated that camptothecin can affect the permeability of the peritrophic membrane to increase its toxicity [63]. Wei et al. studied the synergistic interaction mode of gut microbiota and *B. bassiana* in mosquitoes and confirmed that fungi can reduce the abundance of gut microbiota and the number of probiotics. Fungi can significantly increase the abundance of the opportunistic pathogen *Serratia marcescens,* which overgrows in the midgut and transfers to the hemocoel, causing systemic infections and accelerating mosquito death [60]. Kumar et al. found that the proportion of Proteobacteria increased at 48 h and 96 h but decreased after 144 h after *B. mori* infection with BmBDV. At the genus level, the proportion of *Enterococcus* increased gradually after BmBDV infection of *B. mori*, but the proportion of *Incertae sedis* increased at 96 h, while the proportion of *Lactococcus* decreased at 96 h. *Enterococcus* abundance was positively correlated with the expression levels of *spatzle-1*, *PGRP*-LE, and *PGRP-LB* genes, indicating that the increased abundance of *Enterococcus* activates the Toll and IMD immune pathways [64]. Motta et al. found that glyphosate consumption by honeybees under laboratory or field conditions disrupts the host gut microbiota and affects health [65]. Dai et al. tested the effect of glyphosate on bacterial diversity in the midgut of Italian honeybees in the laboratory. After treatment with 20 mg/L glyphosate, the species diversity and richness in the intestinal tract of honeybees changed significantly, and the survival rate of honeybees decreased [66]. Zhu et al. found that after *Apis mellifera* L. ingested the neonicotinoid insecticide nitenpyram, metabolism, detoxification, and immune-related genes were significantly changed, resulting in an intestinal flora imbalance, which further reduced food consumption and the survival of honey bees [67].

## 4. Negative Effects of Gut Microbes on the Host

Under normal circumstances, the gut microbial community is beneficial to the host. However, the consumption of pathogenic microorganisms, pesticides, and antibiotics by insects will lead to disturbance of the insect gut microbiome and endanger insect health [60,68,69,70]. Gut microbial disturbances refer to changes in the gut community structure far exceeding normal levels that put the host in an unfavorable state. Li et al. found that after *Plutella xylostella* fed on Bt Cry1Ac, gut microbial diversity was significantly reduced, while the relative expression of bacterial load and immune genes was significantly up-regulated, and insect death was accelerated [71]. Tan et al. found that co-infection of *Paranosema locustae* and *B. bassiana* with *Locusta migratoria* altered gut microbiota homeostasis and accelerated death [72]. Previous studies have shown that phoxim exposure to *B. mori* affected gut bacterial community composition and function [69]. Li et al. found that vaccination with sublethal doses of acetamiprid activates the Duox-ROS system and induces ROS accumulation, leading to intestinal dysbiosis and the translocation of *Pseudomonas* and *Staphylococcus* to the hemolymph. Acetamiprid-treated *B. mori* larvae inoculated with *E. cloacae* significantly reduced survival and body weight [73]. The expression of peritrophic membrane protein-related genes *BmCBP-02*, *BmPM-41*, *BmPM-43*, and *BmCDA7,* as well as the toll signaling pathway-related genes *Bmcactus*, *Bmspatzle*, and *Bmrel*, was significantly down-regulated in the gut of *B. mori* following exposure to phoxim, leading to *B. mori’s* susceptibility to *E. cloacae* [74]. Silva et al. found that *G. mellonella* larval mortality and immune responses were significantly increased when inoculated with sub-inhibitory concentrations of nisin and *S. enterica* [75].

In addition, antibiotics are frequently used to prevent bacterial infection of larval honeybees, but prolonged use of antibiotics leads to disruption of gut microbial homeostasis, increasing susceptibility to opportunistic pathogens and affecting honeybee health [76]. Meyel et al. found that after feeding on rifampicin by European earwig larvae, the homeostasis of the host’s gut microbes was disrupted, and the weight of eggs and larvae decreased [77]. Powell et al. found that *Apis mellifera* ingested tylosin tartrate, which significantly increased their susceptibility to *S. marcescens* [78]. Schretter et al. found that *Drosophila* feeding on antibiotics (sterile *Drosophila* larvae) leads to hyperactive motor behaviors. After supplementation with *Lactobacillus brevis*, the motor behavior of *Drosophila* was improved [79]. These results suggest that intestinal microbial homeostasis is crucial to host health.

Some gut commensal bacteria may be transformed into pathogenic bacteria in the presence of dysregulated insect gut microbial homeostasis, such as *S. marcescens*, *Pseudomonas protegens* Ramette, and *B. cereus*, which can elicit host immune responses and disrupt gut microbial homeostasis, and thus pose a serious threat to insect health [80,81,82]. *S. marcescens*, a symbiotic bacteria in the mosquito gut, secretes a secreted protein called SmEnhancin to facilitate arbovirus infection [75,83]. Johnson et al. found that *Drosophila* significantly increased the host’s susceptibility to *B. bassiana* after feeding on *Pseudomonas protegens* Ramette. After *S. exigua* fed on Bt GS57, gut microbial diversity was significantly reduced. Bt GS57 accelerates insect death when complexed with the gut commensal *B. cereus* [81]. To sum up, the interaction between conditioned pathogenic bacteria and insect intestinal microorganisms will affect their insecticidal activity in insects, which provides a new research direction into the pathogenic mechanisms of fungi/bacteria in insects.

## 5. Detection Methods of Insect Gut Microbes

### 5.1. Traditional Identification Methods

The traditional identification methods of gut microbes refer to methods that utilize morphological, physiological, and biochemical characteristics as evaluation criteria. For a long time, the identification of gut microbes has always followed the identification of species by purification culture, morphological, physiological, and biochemical characteristics [84,85,86] (Table 2). At present, the traditional culture detection method needs to inoculate the isolated microorganisms in the NA medium, LB medium, BHI medium, KIA medium, LIA medium, and TSI medium. Due to the limitations of medium components and culture conditions, many insect gut bacteria cannot be cultured [87,88]. Anjum et al. used biochemical analysis, 16S rDNA sequencing, and bioinformatics to identify 150 aerobic or facultative anaerobic bacteria from the guts of 45 worker bees. It was found that there are mainly *Staphylococcus*, *Bacillus*, *Enterococcus*, *Corynebacterium*, and *Micrococcus* in the intestinal tract of honeybees. The isolated bacteria are resistant to acidic environments and ferment sugars, which are beneficial to the survival of bees [89]. Khan et al. used high-throughput sequencing and found that the gut of honeybees contained (Firmicutes, Proteobacteria, Actinobacteria, Flavobacteriia, and Mollicutes) bacteria and fungi (*Dothideomycetes*, *Eurotiomycetes*, *Mucormycotina*, *Saccharomycetes*, *Zygomycetes*, *Yeasts*, and *Molds*) [8]. Broderick et al., using traditional culture methods, found that culturable microorganisms in the gut of *Lymantria dispar* larvae accounted for more than 40% of the bacteria identified by sequencing [9]. Therefore, the types of gut bacteria identified by traditional culture methods are very limited, which has great limitations for revealing the diversity and composition of insect gut microbes.

### 5.2. Electrophoresis

With the development of DNA labeling technology, many new molecular markers have been used for the classification and identification of bacteria, such as restriction/amplified fragment length polymorphism (RFLP/AFLP), denatured gradient gel electrophoresis (DGGE), amplified rDNA restriction analysis (ARDRA), and pulsed-field gel electrophoresis (PFGE). Both RFLP and ARDRA require the use of endonucleases to digest specific PCR amplification products and then obtain fragments of different numbers and lengths to distinguish bacteria. This method is suitable for strains that have been successfully isolated and can determine whether there are differences in the genus of a large number of strains [90,91] (Table 2). He et al. identified 12 and 11 species in the *Camponotus* midgut, respectively, utilizing traditional culture methods and 16S rRNA-RFLP technology [92]. Dec et al. used 16S-ARDRA and MALDI-TOF technology to isolate 80 *Lactobacillus* strains from the cloaca of chicken, goose, and turkey, reaching the species level [93]. AFLP generally requires a pair of endonucleases to digest genomic DNA and then add artificial linkers at both ends of the digested product. Using this as a DNA template, specific primers are used to amplify the DNA. Identification of bacteria by analysis of the different DNA fragments generated by gel electrophoresis [94]. Lindstedt et al. found that AFLP had a better discriminative ability in identifying *C. jejuni* strains than PFGE and PCR-RFLP [95]. DGGE is a technology that separates DNA fragments of the same size with different base compositions, and the melting behavior of the deforming agent at different concentrations is different, resulting in different mobility, so as to separate them [96]. Lin et al. used PCR-DGGE technology to successfully obtain 15 different bands in the study of the microbial diversity in the gut of the diamondback moth *P. xylostella*. Phylogenetic tree analysis revealed that the dominant bacteria in the gut of 4th instar *P. xylostella* larvae belonged to Actinobacteri, Proteobacteria, and Firmicutes [97].

### 5.3. Molecular Detection Method Based on 16S rRNA Gene

At present, microorganisms can be more accurately distinguished according to the variable region in the bacterial 16S rRNA gene sequence, fungal 18S rRNA, or internal transcribed spacer (ITS) sequence. Among them, the 16S rRNA gene is a DNA sequence corresponding to the ribosome 30S small subunit in prokaryotic cells, which is present in all bacterial genomes. The 16S rRNA gene is highly conserved and specific in all bacteria, and universal primers can be used to identify the species of microorganisms. Specific primers or probes can also be designed based on the variable region of the 16S rRNA gene to identify specific strains (Table 2). Snyman et al. collected 78 bacterial strains from the midgut of *Busseola fusca* larvae from 30 sites in South African maize production areas and used 16S rRNA gene sequencing to identify them. The results revealed that the midgut of *B. fusca* larvae mainly contained three phyla, Proteobacteria, Actinobacteria, and Firmicutes, and 20 species from 15 genera, including *Bacillus*, *Enterococcus*, and *Klebsiella* [98]. Yadav et al. collected *Aedes* larvae and pupae in India and used 16S1/16S2 primers to identify the isolated gut bacteria. Twenty-four strains of *Aedes* larval gut bacteria were identified as belonging to four phyla: Proteobacteria, Firmicutes, Bacteroidetes, and Actinobacteria [99]. Based on the 16S rRNA V4 hypervariable region, the 515F/806R primers were used to identify the secondary commensal bacteria *Hamiltonella* in the intestinal tract of *Bemisia tabaci* MED [100]. When the Ham-F/R primers were used to detect the intestinal bacteria of *B. tabaci* by ordinary PCR, the secondary symbiotic bacteria *Hamiltonella* was detected in vivo [101,102]. However, Su et al. used 341F/805R primers to molecularly identify the intestinal microbes in *B. tabaci* MEAM1 but did not discover *Hamiltonella* in its body [103]. However, 16S rRNA amplicon sequencing analysis is usually limited to relative community insights that do not even represent relative abundances well due to extraction and amplification biases. Recent method additions involve the addition of internal and external standards to account for this shortcoming. Therefore, based on the 16S rRNA gene sequence, the preference of the designed primers may lead to different abundances in some bacteria.

### 5.4. qRT-PCR

qRT-PCR is based on the continuous accumulation of PCR reaction products, and the amount of fluorescence increases proportionally. The amount of nucleic acid is determined by detecting the amount of fluorescence in the sample. Regarding qRT-PCR, I consider that the inclusion of controls and independent confirmation of species/strain identification is of the highest importance, e.g., by (high-resolution) melting-curve analysis and sequencing of PCR products. Quantification beyond presence/absence requires standards for each species/strain, but a further improvement is represented by digital PCRs, which really quantify the presence/absence of template molecules (Table 2). Wei et al. used qRT-PCR technology to discover that the relative expression level of *S. marcescens* in the gut of *Anopheles* was significantly up-regulated after *Duox* gene silencing [60]. Tong et al. used qRT-PCR technology to detect *Bacteroides fragilis* in clinical specimens. The results demonstrated that the target bacteria were detected by qRT-PCR from 132 samples (33%), which was much higher than the target bacteria detected by the culture method from 31 samples (8%) [104]. Sedgley et al. used qRT-PCR to detect *E. faecalis* in the provided samples. The results revealed that qRT-PCR detected the target bacteria from five samples (17%), which was much higher than the target bacteria detected from two samples (7%) in the parallel culture experiment [105]. This indicates that qRT-PCR is a rapid and sensitive detection method for identifying bacterial species.
microorganisms-11-01208-t002_Table 2Table 2Detection methods, advantages and disadvantages, and applications of intestinal microorganisms.TechniquesAdvantagesDisadvantagesApplicationReferencesTraditional culture methodLow-cost. Many bacteria cannot be cultured because of the limitations of culture medium and culture condition.*Staphylococcus*, *Bacillus*, *Enterococcus*, *Corynebacterium*, *Micrococcus**Dothideomycetes*, *Eurotiomycetes*, *Mucormycotina*, *Saccharomycetes*, *Zygomycetes*, *Yeasts*, *Molds*[8,87,88,89]ElectrophoresisRapid analysis of a large number of specimens.Suitable for small DNA fragments, and can only reflect the dominant genus.*Lactobacillus, Campylobacter jejuni*[93,95,97]Molecular detection method based on 16S rRNA geneGenerally accurate to the level of genus, a few can be identified to species. For some bacteria, the similarity of sequences in hypervariable regions is very high.Other methods and techniques are needed to identify species with small interspecific differences.*Bacillus*, *Enterococcus*, *Klebsiella,**Hamiltonella*[98,100,101,102]qRT-PCRThe method is simple, rapid, quantifiable, repeatable, sensitive, and specific and can be used to detect live and dead bacteria.High cost, strict requirements for the operation of laboratory personnel, the need for specific instruments and reagents.*Serratia marcescens, Bacteroides fragilis, Enterococcus faecalis*[60,104,105]MetagenomicsMetagenomics overcomes the technical limitations of traditional pure culture methods and can provide information about low abundance or even trace microorganisms in the environment, which can more accurately reflect the true state of microbial survival.The sequencing data are large, the price is much more expensive than 16S sequencing, and the computing resources required for subsequent data processing are high.Major Phylum: Firmicutes, Proteobacteria, Elusimicrobia, Mycobacterium, BacteroidetesMajor Genus: *Enterococcus, Pantoea*, *Acinetobacter*, *Enterobacteriales*, *Lactobacillales*[106,107,108,109,110]

### 5.5. Metagenomics

Metagenomic technology refers to a new technology for studying the structure and function of the microbial community in a sample. Metagenomics can be challenged by biases during extraction, amplification, and subsequent assembly steps (e.g., Gram-positive bacteria with a robust cell wall and species/strains with extremely high/low GC content) (Table 2). The research found that the intestinal microbes of *Termites* are dominated by Firmicutes, Mycobacterium, and Bacteroidetes, and Enterobacteriaceae and Bacteroidetes are the dominant genera [106]. Firmicutes and Proteobacteria are the dominant phyla in the intestinal microbes of the Lepidopteran insects *S. exigua* and *S. litura. S. exigua* is dominated by *Enterococcus* of the family *Enterococcus* [107,109]. *S. litura* is dominated by *Pantoea* and *Acinetobacter* [110]. Insect gut microbes were significantly different from other Lepidopteran insects at the genus level. Therefore, differences at the genus level can be used as an important indicator for evaluating microbial diversity.

## 6. Application of Insect Intestinal Microflora in Pest Management

Commensal bacteria in the insect gut play an important role in enhancing host energy metabolism, immune defense, and resistance to infection by pathogenic microorganisms [14,27,111]. Therefore, in recent years, more and more studies have focused on the function of intestinal flora. It is reported that *B. thuringiensis* kurstaki can induce the mortality of *Lymantri dispar*, which depends on the intestinal bacteria of the host [112]. Paramasiva et al. found that the elimination of intestinal flora in *H. armigera* affected the sensitivity of *B. thuringiensis* to *H. armigera* [113]. However, the insecticidal activity of the *B. thuringiensis* HD-73 strain against *Manduca sexta* and the insecticidal activity of diverse doses of *B. thuringiensis* HD-1 and HD-73 strains against *P. xylostella* larvae indicate that the presence of gut microbes is not essential [114]. Since then, various methods have been used to demonstrate the role of insect gut microbes in the pathogenicity of *B. thuringiensis*, but until now, it has been hard to reach a consistent conclusion.

Insect intestinal symbiotic bacteria have been isolated and purified from different insects, and the functions of a few isolated bacteria have been reported. In mosquitoes, the intestinal symbiotic bacteria *S. marcescens* can significantly promote the tolerance of mosquitoes to arboviruses by secreting toxin proteins [60]. Wei et al. also confirmed that the interaction between *B. bassiana* and mosquito intestinal microbes can accelerate the death of mosquitoes [60]. Ren et al. found that antibiotics secreted by *B. cereus* can decrease the number of symbiotic intestinal flora in the co-infection experiment of *B. cereus* and *B. thuringiensis* and promote the infection of *B. thuringiensis* [115]. Therefore, research on the function of conditionally pathogenic bacteria in the insect intestine may be an important research direction to increase the insecticidal activity of *B. thuringiensis*.

In addition, *B. thuringiensis* can generate Cry toxin protein with insecticidal activity or δ-endotoxin, which, when combined with receptors on intestinal epithelial cells, can induce intestinal epithelial cells to form holes, and intestinal microorganisms will penetrate the blood cavity [116,117,118]. It is reported that *Enterococcus* and *Bacillus*, as the dominant bacteria in the gut of *S. exigua*, are easily able to cause sepsis when they escape to the blood cavity [119]. Mason et al. found that after the larvae of the *M. sexta* were infected with *B. thuringiensis*, *Enterococcus faecalis* would translocate to the blood cavity and induce sepsis, increasing the sensitivity of the *M. sexta* to *B. thuringiensis* [120]. These results indicate that the translocation of intestinal symbiotic bacteria to the hemocoel is an important factor causing insect death.

## 7. Conclusions

Insect guts are inhabited by a large number of microorganisms and are rich in microbial resources. The gut microbiota is essential for host nutrient metabolism, immune defense, and drug resistance, and trials using insects as models have revealed that changes in microbiota structure can affect host health (Figure 1). Intestinal microbes also include viruses, bacteriophages, eukaryotic microbes, etc. The functional studies of these microbes are not yet perfect, and further in-depth research is needed. In addition, the details of the interaction between the insect immune system and the gut microbiota still need to be further explored.

At present, important progress has been made in the study of the taxonomic composition of insect intestinal microorganisms and their effects on growth and development. However, there are abundant microorganisms on the surface of insect skin, feces, and living environments, which can also affect the health of insects. The relationship between insect growth environment and host intestinal flora requires further exploration. Therefore, the research in this field will assist us in better understanding how to control intestinal flora according to the living environment, diet, and physiological changes of insects and will further promote our understanding of insect intestinal microbial functions, thus providing new opportunities for enhancing the overall health of beneficial insects and the management of harmful insects.

## Figures and Tables

**Figure 1 microorganisms-11-01208-f001:**
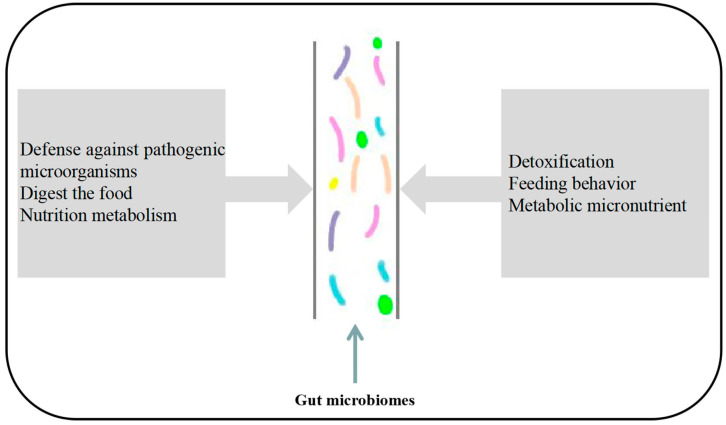
The main function of gut microbiota in insects.

**Table 1 microorganisms-11-01208-t001:** The role of insect gut microbes in nutrient metabolism.

Insect	Bacteria Genera	Function	References
*Bombyx mori*	*Pseudomonas*, *Klebsiella pneumoniae*, *Clostridium flexneri*	It secretes cellulase, which breaks down carbohydrates to provide energy for the host.	[14]
*Locusta migratoria manilensis*	*Klebsiella pneumoniae*	It breaks down grass into carbohydrates, amino acids and sugars, providing energy to the host.	[15]
*Aedes albopictus*	*Lelliottia*, *Cladosporium*, *Aspergillus*, *Ampullimonas*, *Cyberlindnera*	It can digest food and provide nutrients for insects.	[16]
Herbivorous turtle ants *Cephalotes*	*Burkholderiales*, *Opitutales*	It can participate in nitrogen recycling to provide amino acids for the host.	[13]
*Acyrthosiphon pisum*	*Buchnera aphidicola*	It can provide the host with vitamins B2 and B5.	[6]
*Honeybee*	*Gilliamella apicola* and *Lactobacillus* sp.	It produces SCFAs that promote host growth.	[19]
*Drosophila*	*Acetobacter pomorum*, *Lactobacillus plantarum*, *Saccharomyces cerevisiae*, *Lactobacillus plantarum*, *Acetobacter malorum*	It can influence the feeding behavior of the host.	[23]
*Rhynchophorus ferrugineus* Olivier	*Lactococcus lactis*, *Enterobacter cloacae*	It can affect protein, glucose, and triglyceride levels in the host’s hemolymph.	[25]

## Data Availability

Not applicable.

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
