# Peer review of "Research Progresses on the Function and Detection Methods of Insect Gut Microbes"

_microorganisms, 2023, doi:10.3390/microorganisms11051208_

Round 1

Reviewer 1 Report

This review article intended to discuss the importance of microbiota in the gut of insects as their diversity significantly affects the health of insects. The authors tried to throw light on the functions, different factors that influence their diversity and detection methods of microbes in insect gut. This article looks good and authors albeit reviewed the literature related to gut microbiota but I noticed the authors did not cited many of the articles published on the role of diversity and functions of microbiota in insects – e.g., Zhang et al., 2022 (Microorganisms, 10(6): 1234; https://doi.org/10.3390/microorganisms10061234), this article and others on this topic should be thoroughly discussed in ms.

I noticed that some of the headings and sub-headings are same to this article, the authors should distinguish their work from other published literature on this topic and the novelty in the work should be highlighted.

The Introduction section should be further elaborated with more citations and pictorial presentation of different aspects should be added in the article as this will attract more readerships.

The authors should seek help from English native colleague to read the ms for fluency in language as in this form it does not commensurate with this highly reputed journal.

I am of the opinion that this review article cannot be accepted in current form for its publication in Microorganisms – however, The Editor may allow authors to significantly improve the article and re-submit to be considered for its publication.

Author Response

Dear Reviewers:

Thank you very much for your letter and the comments about our paper entitled “Research progresses on the function and detection methods of insect gut microbes” (ID: 2290514). The comments are valuable and helpful for revising and improving our paper, as well as the important guiding significance to our research. We have carefully taken the comments to make the paper clearer and more compelling. The point to point responses to the editors are list as following. Thanks for all the help.

Please, provide the following corrections:

- The authors did not cited many of the articles published on the role of diversity and functions of microbiota in insects–e.g., Zhang et al., 2022 (Microorganisms, 10 (6): 1234; https://doi.org/10.3390/microorganisms10061234), this article and others on this topic should be thoroughly discussed in ms.

Response: Thank you for your advice. We have added new relevant literature. Please refer to literature in the manuscript [2], [7], [11], [22], [28], [37] and [55]. 

- I noticed that some of the headings and sub-headings are same to this article, the authors should distinguish their work from other published literature on this topic and the novelty in the work should be highlighted.

Response: Thanks for your suggestion. The headings and sub-headings of the manuscript have been changed (the text is marked in red) .

-The Introduction section should be further elaborated with more citations and pictorial presentation of different aspects should be added in the article as this will attract more readerships.

Response: Thanks for your valuable advice. The introduction has been elaborated further. Please refer to the introduction section. Please refer to the introduction section part L24-L36.

- The authors should seek help from English native colleague to read the ms for fluency in language as in this form it does not commensurate with this highly reputed journal.

Response: Thanks for your valuable advice. We tried our best to improve the manuscript and made some changes to the manuscript. These changes will not influence the content and framework of the paper. And here we did not list the changes but marked in red in the revised paper. We appreciate for Reviewers' warm work earnestly and hope that the correction will meet with approval.

Reviewer 2 Report

Manuscript: microorganisms-2290514 - Research progresses on the function and detection methods of insect gut microbes

The review's authors found that the insect guts are inhabited by a large number of microorganisms and are rich in microbial resources. The gut microbiota is critical for host nutrient metabolism, immune defense, and drug resistance, and experiments using insects as models have demonstrated that changes in microbiota structure can affect host health. The details of the interaction between the insect immune system and the gut microbiota still need to be further explored. In recent years, the development and application of new technologies such as metagenomics and bioinformatics have greatly improved the understanding of the composition and diversity of the insect gut microbiota. This review summarizes insect gut function and the factors that affect insect gut microbes. Also, The authors compared past and present methods for detecting insect intestinal microorganisms.

The literature analysis methods are correct.

The English of the text is well written and well readable but needs additional checking with a professional translator.

The uniqueness of the text is more than 90% by AntiPlagiarism.NET.

The text contains some misspellings and typos. Also need to expand the part of the discussion.

There are some comments and questions:

1) Line 37 - in thegut - should be - in the gut.

2) Line 140 - After the sentence - Trinder et al. found that supplementation of Lactobacillus rhamnosus in Drosophila melanogaster diet could reduce the toxicity of organophosphorus pesticides [30]. - add additional sentence - Danilenko et al. described that Lactobacillus species of L. plantarum (strains H28, H24,  KX519413, KX519414, LP8, LP25, LP86, LP95, LP100) and L. kunkeei were characterized by an increased antioxidant  potential that protects against different pesticides in the gut of honey bees (Danilenko et al., 2021).

3) Add to the References: Danilenko, V.N.; Devyatkin, A.V.; Marsova, M.V.; Shibilova, M.U.; Ilyasov, R.A.; Shmyrev, V.I. Common inflammatory mechanisms in COVID-19 and Parkinson’s diseases: the role of microbiome, pharmabiotics and postbiotics in their prevention. J Inflamm Res 2021, 14, 6349–6381, doi:10.2147/JIR.S333887.

4) Line 355 -  spiecies/strain - should be - species/strain.

5) Line 466 - theAedes albopictus - should be - the Aedes albopictus.

6) Please add chapter about antioxidant function of insect gut microbiota.

Please improve the manuscript according to the above comments.

Author Response

Dear Reviewers:

Thank you very much for your letter and the comments about our paper entitled “Research progresses on the function and detection methods of insect gut microbes” (ID: 2290514). The comments are valuable and helpful for revising and improving our paper, as well as the important guiding significance to our research. We have carefully taken the comments to make the paper clearer and more compelling. The point to point responses to the editors are list as following. Thanks for all the help.

Please, provide the following corrections:

- The English of the text is well written and well readable but needs additional checking with a professional translator. The text contains some misspellings and typos. Also need to expand the part of the discussion.

Response: Thanks for your valuable advice. We tried our best to improve the manuscript and made some changes to the manuscript. These changes will not influence the content and framework of the paper. And here we did not list the changes but marked in red in the revised paper. We appreciate for Reviewers' warm work earnestly and hope that the correction will meet with approval.

- Line 37 - in thegut - should be - in the gut.

Response: Thank you for your advice. Changes have been made in the revised manuscript. Please refer to the 2. Functional roles of the insect gut Microbiota section part L47.

- Line 140 - After the sentence - Trinder et al. found that supplementation of Lactobacillus rhamnosus in Drosophila melanogaster diet could reduce the toxicity of organophosphorus pesticides [30]. - add additional sentence - Danilenko et al. described that Lactobacillus species of L. plantarum (strains H28, H24,  KX519413, KX519414, LP8, LP25, LP86, LP95, LP100) and L. kunkeei were characterized by an increased antioxidant potential that protects against different pesticides in the gut of honey bees (Danilenko et al., 2021).

Response: Thanks for your valuable advice. Changes have been made in the revised manuscript. Please refer to the 2.4 Enhance host drug resistance section part L177-L181. 

- Add to the References: Danilenko, V.N.; Devyatkin, A.V.; Marsova, M.V.; Shibilova, M.U.; Ilyasov, R.A.; Shmyrev, V.I. Common inflammatory mechanisms in COVID-19 and Parkinson's diseases: the role of microbiome, pharmabiotics and postbiotics in their prevention. J Inflamm Res 2021, 14, 6349–6381, doi:10.2147/JIR.S333887.

Response: Changes have been made in the revised manuscript. Please refer to the References section part L593-L595. 

- Line 355 -  spiecies/strain - should be - species/strain.

Response: Changes have been made in the revised manuscript. Please refer to the 5.5. Metagenomics section part L425. 

- Line 466 - theAedes albopictus - should be - the Aedes albopictus.

Response: Changes have been made in the revised manuscript. Please refer to the References section part L536.

- Please add chapter about antioxidant function of insect gut microbiota.

Response: Changes have been made in the revised manuscript. Please refer to the 2.3. Antioxidation function section part L149-L166.

Reviewer 3 Report

The aim of the present manuscript is to introduce the subject and furnish the reader basic knowledge on insect microbiome. This manuscript is mostly a permutation of already published and reviewed literature. Nevertheless, as few review articles appeared with the aim to re-structure our knowledge and to give a particular direction for future studies, the addition of the results published in the past three years is a positive increment.  The literature is fairly cited, slightly biased but lack references for basic review articles, as https://doi.org/10.1371/journal.ppat.1008398 and doi: 10.3389/fmicb.2020.01357. This article, if published, will provide basic knowledge for understanding the importance of gut microbiome.

Author Response

Dear Reviewers:

Thank you very much for your letter and the comments about our paper entitled “Research progresses on the function and detection methods of insect gut microbes” (ID: 2290514). The comments are valuable and helpful for revising and improving our paper, as well as the important guiding significance to our research. We have carefully taken the comments to make the paper clearer and more compelling. The point to point responses to the editors are list as following. Thanks for all the help.

Please, provide the following corrections:

- The literature is fairly cited, slightly biased but lack references for basic review articles, as https://doi.org/10.1371/journal.ppat.1008398 and doi: 10.3389/fmicb.2020.01357.

Response: Thank you for your advice. We have added new review articles. Please refer to literature in the manuscript [2], [7], [11], [22], [28], [37] and [55]. 
